# First Report of the Molecular Mechanism of Resistance to Tribenuron-Methyl in *Silene conoidea* L.

**DOI:** 10.3390/plants11223044

**Published:** 2022-11-10

**Authors:** Ying Sun, Yujun Han, Hong Ma, Shouhui Wei, Yuning Lan, Yi Cao, Hongjuan Huang, Zhaofeng Huang

**Affiliations:** 1State Key Laboratory for Biology of Plant Diseases and Insect Pests, Institute of Plant Protection, Chinese Academy of Agricultural Sciences, Beijing 100193, China; 2College of Agriculture, Northeast Agricultural University, Harbin 150030, China

**Keywords:** *Silene conoidea* L., acetolactate synthase (ALS), target-site-based resistance, metabolic resistance

## Abstract

*Silene conoidea* L. is an annual troublesome broadleaf weed in winter wheat fields in China. In recent years, field applications of tribenuron-methyl have been ineffective in controlling *S. conoidea* in Hebei Province, China. The aim of this study was to determine the molecular basis of tribenuron-methyl resistance in *S. conoidea*. Whole-plant response assays revealed that the resistant population (R) exhibited a higher level of resistance (382.3-fold) to tribenuron-methyl. The R population also showed high cross-resistance to other acetolactate synthase (ALS) inhibitors, including imazethapyr, bispyribac-sodium and florasulam. However, the R population could be controlled by the field-recommended rates of bentazone, MCPA, fluroxypyr, carfentrazone-ethyl and bromoxynil. In vitro ALS activity assays indicated that the tribenuron-methyl I_50_ value for the R population was 18.5 times higher than those for the susceptible population (S). ALS gene sequencing revealed an amino acid mutation, Trp-574-Leu, in the R population. Pretreatment with the P450 inhibitor malathion indicated that the R population might have cytochrome P450-mediated metabolic resistance. These results suggest that the Trp-574-Leu mutation and P450-mediated enhanced metabolism coexist in *S. conoidea* to generate tribenuron-methyl resistance. This is the first time that target-site and non-target-site resistance to tribenuron-methyl has been reported in *S. conoidea*.

## 1. Introduction

*Silene conoidea* L. is found in subtropical and temperate regions of western Eurasia, as well as parts of the Himalayas. It is an annual broadleaf weed that can cause damage to wheat, barley, lentil pea, mustard and roquette crops [1]. It is widely distributed in the winter wheat cropping regions in China. The seeds of *S. conoidea* may fall on the soil surface or be mixed with wheat seeds during the wheat harvest [2]. After seedling emergence, it can compete with wheat for water, fertilizer, sunlight and space, lowering wheat yields [1]. Previous research has shown that *S. conoidea* occurs with high frequency in the field and has become the dominant weed in wheat fields in Hebei Province, thus resulting in a severe impact on local wheat production [3]. Moreover, this plant is likely to become one of the most threatening weed species in wheat fields because of its high reproductive capacity, strong adaptability and rapid spread.

Herbicides have been the most important weed management approach due to their high efficiency and quicker action [4]. Herbicide resistance is the result of weed evolutionary adaptation to herbicide selection, and it can be divided into two types: target-site-based resistance (TSR) and non-target-site-based resistance (NTSR) [5]. TSR is provided by amino acid substitutions in the herbicide target protein, which can modify the structure or function of target enzymes, or by the overexpression of target genes [6,7]. To date, nine mutation sites have been reported to confer target-site resistance to ALS inhibitors: Ala-122, Pro-197, Ala-205, Phe-206, Asp-376, Arg-377, Trp-574, Ser-653 and Gly-654 [8,9]. NTSR is achieved through reduced absorption or translocation, as well as enhanced sequestration or metabolic degradation [10]. In comparison to TSR, NTSR is more complex and frequently involves several enzyme superfamilies, such as cytochrome P450s (P450s), glucosyltransferases (GTs), glutathione S-transferases (GSTs), aldo-keto reductases, transporters and esterases [11,12,13]. Furthermore, TSR and NTSR can coevolve in response to herbicide selective pressure; hence, the two systems may coexist in the same species, population, or individual [14,15,16].

Tribenuron-methyl (methyl 2-[[[[N-(4-methoxy-6-methyl-1,3,5triazin-2-yl)methyl-amino]carbonyl]amino]sulfonyl]benzoate) is one of the most popular acetolactate synthase (ALS) inhibitors used in wheat fields. It belongs to sulfonylureas (SUs), which have five structurally distinct chemical groups, including imidazolinones (IMIs), triazolopyrimidines (TPs), pyrimidinylthio-benzoates (PTBs) and sulfonylamino-carbonyltriazolinones (SCTs). Due to the advantages of broad spectrum, low dosage and safety for after reap crops, this chemical has become the main herbicide used for annual and perennial broadleaf weed control in wheat fields with the largest application area and the longest application duration in China [17]. Previous studies have found that tribenuron-methyl was more than 95% effective in the control of *S. conoidea* in winter wheat fields in the provinces of Hebei and Shandong [18,19]. However, some regions of China have failed to control *S. conoidea* with the field-recommended dose of tribenuron-methyl in recent years. At present, 48 weeds worldwide have resistance to tribenuron-methyl [9], including *Descurainia sophia* [20], *Capsella bursa-pastoris* [21] and *Myosoton aquaticum* [22]. However, the resistance mechanisms of *S. conoidea* against tribenuron-methyl, or even other herbicides, have not yet been characterized.

It is important to establish efficient and viable weed management strategies to clarify the resistance mechanisms of *S. conoidea* to tribenuron-methyl. Therefore, the objectives of this study are to (1) evaluate the resistance level of *S. conoidea* to tribenuron-methyl; (2) investigate the potential resistance mechanisms involved in tribenuron-methyl; and (3) assess the sensitivity of *S. conoidea* to other common herbicides.

## 2. Results

### 2.1. Dose—Response Assay

The whole-plant dose response was used to determine the resistance level of the R population to tribenuron-methyl. When treated with the field dose (22.5 g a.i. ha^−1^), the S population was 100% controlled; however, tribenuron-methyl rates up to 360 g a.i. ha^−1^, which is 16-fold the recommended dose, did not completely extirpate the R population (Figure 1). The fresh weight of the S population declined by 50% in reaction to a tribenuron-methyl rate of 0.47 g a.i. ha^−1^, while the GR_50_ of the R population occurred in response to 179.70 g a.i. ha^−1^ (Table 1). The R population showed higher resistance (resistance factor, RI = 382.3) as compared to the S population.

### 2.2. In Vitro ALS Activity Assay

In the absence of tribenuron-methyl, the total ALS activities were 28.22 and 35.45 nmol acetoin mg^−1^ protein min^−1^ for the S and R populations, respectively (Table 2). The data obtained revealed that the enzyme activity of ALS from the R population was much more insensitive to tribenuron-methyl than that of the S population. As shown in Figure 2, the I_50_ value was 214.95 μM for the R population. However, for the S population, the I_50_ value was 11.61 μM. The I_50_ value for the R population was 18.5-fold greater than that of the S population.

### 2.3. Identification of ALS Mutations in S. conoidea

The PCR fragments of *S. conoidea* covering the nine verified mutation sites were amplified from the S and R populations. These gene sequences were trimmed to a size of 1804 bp. The obtained sequence of *S. conoidea* was submitted to GenBank (NCBI) (accession number: ON152802). The single nucleotide replacement in the ALS gene of the R population resulted in an amino acid substitution of tryptophan (TGG) for isoleucine (TTG) at position 574 compared with the S population (Figure 3).

### 2.4. Effect of Malathion on Tribenuron-Methyl Resistance

To investigate the potential for herbicide metabolism within the R population, the fresh weight reduction rate was determined. The fresh weight of the R population decreased by 94.6% at a dose of 90 g a.i. ha^−1^ compared to 25.4% for the plants pretreated with 1000 g a.i. ha^−1^ malathion. However, the fresh weight of the S population did not change significantly in response to tribenuron-methyl plus malathion pretreatment (Table 3).

### 2.5. Herbicide Cross-Resistance and Multiple Resistance

The resistance pattern of *S. conoidea* to the other herbicides was also determined in this study. As shown in Table 4, the results of the whole-plant response experiments indicated that the R population displayed high levels of cross-resistance to other ALS-inhibiting herbicides, such as imazethapyr (946.1-fold), bispyribac-sodium (80.8-fold) and florasulam (>1787-fold). However, both R and S populations were fully managed by bentazone, MCPA, fluroxypyr, carfentrazone-ethyl and bromoxynil at their recommended field doses.

## 3. Discussion

Herbicides have dominated weed control practices in China since their introduction in the late 1950s, and herbicide resistance has steadily increased over the last decade [23]. The SU herbicide tribenuron-methyl has been constantly used in winter wheat fields in China since 1988 [24]. Weed resistance to the herbicide tribenuron-methyl may progressively develop with increasing years of application.

Resistance to tribenuron-methyl (382.3-fold) was confirmed in *S. conoidea* in this study. To characterize the molecular mechanisms of resistance, we sequenced the ALS gene and found the Trp-574-Leu mutation present in the R population. The Trp-574-Leu mutation has been identified in 38 weed species worldwide [25], including *Amaranthus retroflexus* [26], *Echinochloa crus-galli* [27] and *Descurainia Sophia* [28]. Previous research reported that Trp574 mutation usually shows broad-spectrum resistance to all five classes of ALS inhibitors [29]. The R population of *S. conoidea* with the Trp574 mutation had higher cross resistance to other ALS inhibitors, such as imazethapyr (IMI), bispyribac-sodium (PTB) and florasulam (TP) in the study. Therefore, Trp-574-Leu was mainly responsible for *S. conoidea* resistance to tribenuron-methyl.

Some amino acid residues have been found to be important for both binding to herbicides and maintaining ALS catalytic activity, so a mutation at those residues inevitably results in the loss of many contacts with herbicides [28,30]. In this study, the I_50_ value for the R population was 18.5-fold higher than that for the S population. Our results are similar to the results reported by Liu et al. [22], who found that the resistant flixweed population showed decreased sensitivity to tribenuron-methyl in ALS activity experiments. Additionally, despite this trend being consistent with our dose–response analysis, the RI from the in vitro ALS activity assay was much lower in comparison to the whole-plant dose response. This result suggests that other herbicide-resistance mechanisms, such as improved herbicide metabolism, likely play a role in the resistance to tribenuron-methyl in the R population.

Malathion, a P450 inhibitor, has been used to identify metabolic resistance involving P450s in various weeds, including *Alopecurus aequalis* [31], *Echinochloa phyllopogon* [32] and *Descurainia Sophia* [33]. Malathion could effectively decrease fresh weight when it was applied with the herbicide in the current study, indicating the P450s may be involved in the resistance to tribenuron-methyl in *S. conoidea*. However, this result only indirectly demonstrated the participation of one or more P450s in this resistance, but failed to offer any information regarding the specific P450s involved. Most evidence of P450 involvement in herbicide resistance was obtained by RNA-seq, which is currently the most effective method to identify genes differentially regulated among experimental modalities [34,35,36]. Further research will focus on identifying candidate genes involved in non-target-site-based tribenuron-methyl resistance in *S. conoidea*.

In summary, the evolution of target-site resistance and possibly non-target-site resistance mechanisms in the resistant *S. conoidea* population suggest a serious threat to the sustainable use of the ALS inhibitors for controlling this weed in wheat fields. Fortunately, the *S. conoidea* population could be controlled by the field-recommended rates of bentazone, MCPA, fluroxypyr, carfentrazone-ethyl and bromoxynil. Herbicide rotation with different modes of action is a useful measure for delaying the evolution of resistance in *S. conoidea*. In addition, non-target-site resistance, such as enhanced metabolism, requires further study, because it may confer resistance to herbicides with completely different modes of action. This affects the effectiveness of herbicide rotation in slowing the evolution of resistance [37].

## 4. Materials and Methods

### 4.1. Plant Materials and Growth Conditions

Seeds from *S. conoidea* were collected from Hebei Province, China, in June 2021. The mature seeds of the susceptible population (S) were obtained from a distant area of Xingtai, Hebei Province (E114°30′25″, N37°4′4″) that had never been treated with herbicides. Resistant population (R) seeds were collected from wheat fields in Dingzhou, Hebei Province (E115°4′6″, N38°37′21″), with a history of more than 10 successive years of tribenuron-methyl use. Both groups of *S. conoidea* seeds were air-dried in the shade, stored in paper bags until needed, and then soaked overnight in 0.4% gibberellic solution at 4 °C before seeding. A total of 20 seeds from each S and R population were planted in 7-cm radius pots containing a 1:1 (*v*/*v*) peat:sand mixture. Plants were cultured in a greenhouse (30/25 °C and 16/8 h of day/night) at the Institute of Plant Protection, Chinese Academy of Agricultural Sciences, with regular watering and fertilization. Seedlings were thinned to 6 plants per pot when they reached the three- to four-leaf stage.

### 4.2. Whole-Plant Dose Response

Tribenuron-methyl was treated at the three- to four-leaf stage using a laboratory sprayer with a flat-fan nozzle, providing 30 L ha^−1^ at 220 kPa. Tribenuron-methyl was applied at 0, 90, 180, 360, 720, 1440 and 2880 g of active ingredient (a.i.) ha^−1^ to the R population, and at 0, 0.175, 0.35, 0.46, 0.70, 1.40 and 2.81 g a.i. ha^−1^ to the S population. The recommended rate for tribenuron-methyl is 22.5 g a.i. ha^−1^. After herbicide applications, the seedlings were cultured in the greenhouse. The herbicide control effect was assessed by weighing the fresh weight of plants 21 days after treatment (DAT). This experiment was conducted twice in a completely randomized design with three replications.

### 4.3. In Vitro Enzyme Activity Assay

Fresh leaves from both the S and R populations were collected at the three- to four-leaf stage and stored in liquid nitrogen at −80 °C for ALS extraction. A protein concentration of the extract was quantified using the Bradford method [38]. The ALS activity assays were conducted according to the methods described by Yu et al. [39]. Enzyme activity was colorimetrically determined (530 nm) by measuring the amount of acetoin formed. The herbicide concentrations for tribenuron-methyl used for enzyme activity testing were 0, 2.47, 4.94, 9.88, 19.76, 39.52, 79.04, 158.07 and 316.14 μM for the S population, and 0, 9.88, 19.76, 39.52, 79.04, 158.07, 316.14, 632.29 and 1264.57 μM for the R population. The herbicide concentration required to lower enzyme activity by 50% compared to the untreated control was calculated as I_50_. Three subsamples from each extraction were assayed, and two extractions per population were used.

### 4.4. Sequencing of the ALS Gene

The young shoot tissue from 10 surviving plants of each population treated with tribenuron-methyl at the three-leaf stage was harvested, immediately frozen in liquid nitrogen, and stored at −80 °C until use. DNA was extracted from young shoot tissue using a Plant Genomic DNA Kit (Tiangen Biotech, Beijing, China) according to the manufacturer’s instructions.

To identify the molecular basis for resistance, ALS gene fragments, including all nine site mutations conferring ALS resistance, were amplified, sequenced and compared. The corresponding primers (shown in Table 5) were designed according to the sequences of *Myosoton aquaticum* (KF589890.1) using Primer Premier 5.0 software. PCR was performed in a 20-μL reaction mixture containing 1 μL of genomic DNA, 0.5 μL each of both forward and reverse primers, 10 μL of Taq 2× master mix and 8 μL of ddH_2_O (Tiangen Biotech, Beijing, China). PCR was performed as follows: 3 min incubation at 94 °C; 35 cycles of 30 s at 94 °C, 30 s at 58 °C, and 1 min at 72 °C; and then 5 min at 72 °C, where X was the annealing temperature for each primer pair used. The PCR products from the *ALS* gene were sequenced by Biomed Biotech (Beijing, China) after gel purification and compared using Vector NTI 12.5 (SigmaPlot Software Inc., Chicago, IL, USA).

### 4.5. Effect of Malathion on Tribenuron-Methyl Resistance

Malathion (1,2-di(ethoxycarbonyl)ethyl o,o-dimethylphosphorodithioate) is a common indicator for P450-mediated metabolic resistance to ALS-inhibiting herbicides. At the three- to four-leaf stage, *S. conoidea* was treated with tribenuron-methyl in the absence and presence of malathion. Malathion at a concentration of 1000 mg L^−1^ was applied to the S and R populations before tribenuron-methyl application to evaluate the effect of metabolic resistance. The doses of tribenuron-methyl used were 90 g a.i. ha^−1^ for the R population and 0.35 g a.i. ha^−1^ for the S population. The aboveground shoots were harvested 21 days after treatment, and the fresh weight was recorded.

### 4.6. Sensitivity to Other ALS Inhibitors and Alternative Herbicides

To find other applicable herbicides to control the R population of *S. conoidea*, assays were conducted to determine the cross-resistance and multiple resistance profiles. The herbicide treatments are listed in Table 6. The herbicide treatments were performed in the same method as described in the whole-plant dose-response assay. The control effort was assessed by calculating the death rate at 21 days after treatment (DAT), which was defined as no new healthy leaves being generated.

### 4.7. Statistical Analysis

The herbicide dose–response assays were repeated three times with three replications each time. The significance of the regression parameters was determined using a t test (*p* = 0.05). When there was no significant difference between the results of each experiment, the data were pooled. The data were combined and fitted to a nonlinear regression analysis in SigmaPlot v.12.0 (Systat Software, Inc., San Jose, CA, USA). The GR_50_ was determined by using a four-parameter log-logistic equation (Seefeldt et al., 1995). The fitted model was as follows:$$Y=C+\frac{D-C}{\left[1+{\left(\frac{X}{{\mathrm{GR}}_{50}}\right)}^{b}\right]}$$

C is the lower limit of the response, D is the upper limit of the response, X is the herbicide application rate, and b is the slope of the curve at GR_50_.

To estimate the resistance levels, the resistance index (RI) was calculated as follows:$$\mathrm{RI}=\frac{{\mathrm{GR}}_{50}\left(R\right)}{{\mathrm{GR}}_{50}\left(S\right)}$$

## 5. Conclusions

This study demonstrates, for the first time, the coexistence of target-site-based (Trp-574-Leu mutation) and possibly non-target-site-based (improved herbicide metabolism) resistance to tribenuron-methyl in *S. conoidea* populations from China. Given the evolution and incidence of TSR and NTSR in the resistant *S. conoidea* population, increasing tribenuron-methyl rates would not be an effective strategy for reducing the resistant population. Moreover, Trp574 mutation exhibited broad-spectrum resistance to other classes of ALS inhibitors in this study, which also might make controlling this population difficult. To manage this resistant weed, other modes of herbicides commonly used in crop fields for the control of broadleaf weeds, such as bentazone, MCPA (2-methyl-4-chlorophenoxyacetic acid), fluroxypyr, carfentrazone-ethyl and bromoxynil, should be used.

## Figures and Tables

**Figure 1 plants-11-03044-f001:**
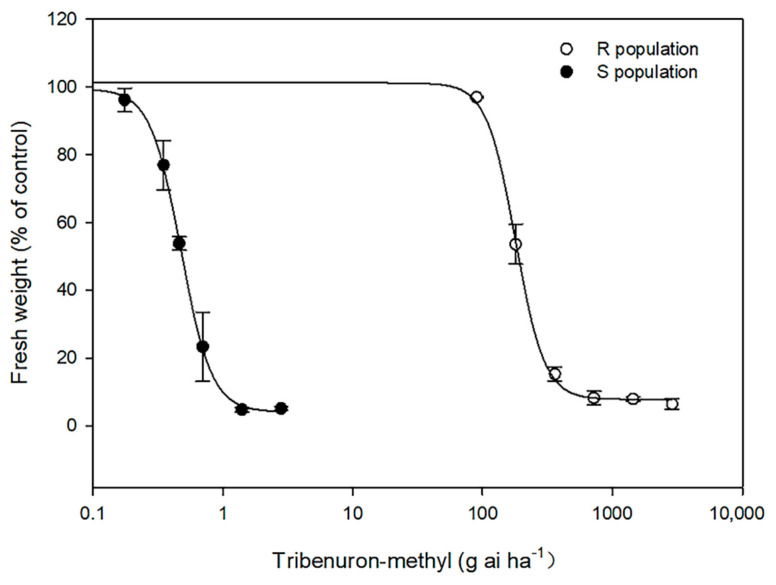
Dose-response curve for the R(○) and S(●) populations treated with different concentrations of tribenuron-methyl. Each point represents the mean of two experiments, each containing three replicates, and error bars represent the standard error. The fresh weight reduction rate is presented as a percentage of that recorded from the untreated plants.

**Figure 2 plants-11-03044-f002:**
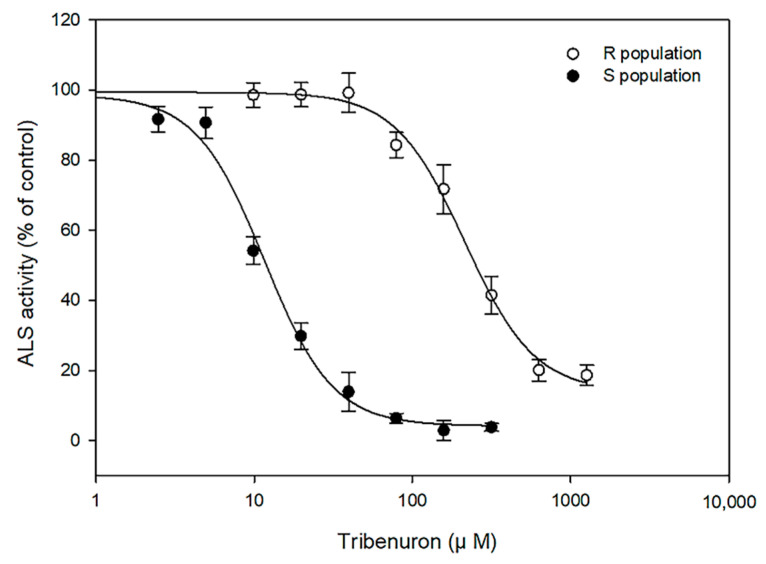
Tribenuron-methyl inhibition of ALS in isolates from the R(○) and S(●) populations of *Silene conoidea*. ALS activity is expressed as the percentage of activity in the absence of tribenuron-methyl. Each point indicates the average of two experiments that each included three replicates. Vertical bars represent the standard error.

**Figure 3 plants-11-03044-f003:**
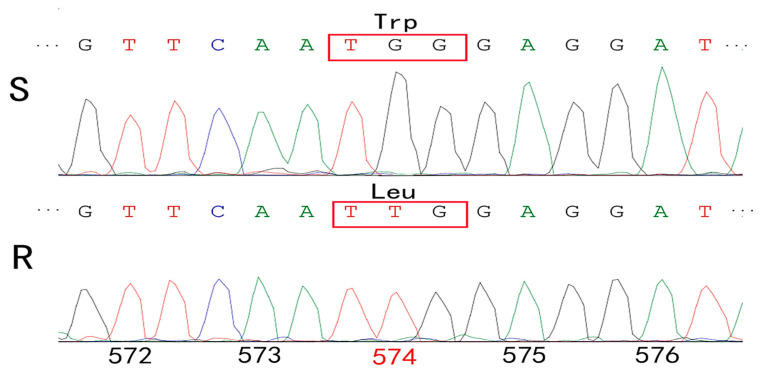
Partial ALS gene sequencing results showing a Trp-574-Leu mutation resulting from a TGG to TTG change in the R population. The amino acid position of ALS was based on *Arabidopsis*.

**Table 1 plants-11-03044-t001:** GR_50_ values of S and R populations of *Silene conoidea* treated with tribenuron-methyl. Each value represents the mean (SE) of two experiments, each containing three replicates. RI was calculated as the ratio of GR_50_ values from resistant and susceptible populations.

Population ^a^	GR_50_ (g a.i. ha^−1^) ^b^	RI ^c^
S	0.47 ± 0.01	-
R	179.70 ± 4.01	382.3

^a^ S, susceptible; R, resistant. ^b^ GR_50_, herbicide rate causing a 50% growth reduction of plants. ^c^ RI, GR_50_ value of the R population/GR_50_ value of the S population.

**Table 2 plants-11-03044-t002:** In vitro ALS activity of R and S populations of *Silene conoidea*. The values represent the mean ± SE of two experiments, each containing three replicates.

Population ^a^	Total ALS Activity (nmol Acetoin mg^−1^ Protein min^−1^)	R/S ^b^	I_50_ (μM) ^c^	R/S ^b^
S	28.22 ± 2.37	-	11.61 ± 1.00	-
R	35.45 ± 1.72	1.3	214.95 ± 18.59	18.5

^a^ S, susceptible; R, resistant. ^b^ I_50_ refers to the herbicide concentration required to inhibit ALS activity by 50% compared with the untreated control. ^c^ The resistance ratio (R/S) was calculated by dividing the I_50_ value of the resistant population by that of the susceptible population.

**Table 3 plants-11-03044-t003:** Fresh weight control rate of susceptible (S) and resistant (R) populations of *Silene conoidea* in response to tribenuron-methyl with or without malathion pretreatment.

Population ^a^	Application Dose (g a.i. ha^−1^)	Fresh Weight Reduction (%) ^b^	
		Tribenuron-Methyl	Tribenuron-Methyl + Malathion
S	0.35	76.9	74.5
R	90	94.6	25.4

**^a^** S, susceptible; R, resistant. **^b^** Reduction rate, the ratio of fresh weight to that of the control group.

**Table 4 plants-11-03044-t004:** GR_50_ values of the *Silene conoidea* populations treated with ALS inhibitors and the resistance index (RI) indicated by the R/S ratio.

Herbicide	Population ^a^	GR_50_ (g a.i. ha-1) ^b^	RI ^c^
Imazethapyr	S	0.88 ± 0.06	-
	R	832.53 ± 238.31	946.1
Bispyribac-sodium	S	0.21 ± 0.01	-
	R	16.97 ± 0.77	80.8
Florasulam	S	<0.04	-
	R	71.48 ± 0.02	>1787

^a^ S, susceptible; R, resistant. ^b^ GR_50_, herbicide rate causing 50% plant mortality. Each value represents the mean ± standard error. ^c^ RI, GR_50_ value of the R population/GR_50_ value of the S population.

**Table 5 plants-11-03044-t005:** Primers used in this study.

Primers	Sequence (5′-3′)	Amplicon Size (bp)	Annealing Temperature (°C)	Amplification Point
ALS-1F	5′-CGCCGCAAATACCAAAACCACTCCC-3′	649	58.5	122, 197
ALS-1R	5′-CCACCACCAACATACAGAACA-3′			
ALS-2F	5′-CAAGTTCCGAGGCGAATGAT-3′	894	56	205, 206, 376, 377
ALS-2R	5′-CAAGCCCACTGGAGGTCA-3′			
ALS-3F	5′-CGAGGGTGAGGAAGAGCA-3′	675	55	574, 653, 654
ALS-3R	5′-TCTTCCATCACCCTCGTTTA-3′			

**Table 6 plants-11-03044-t006:** Information on herbicides used in cross resistance and multiple resistance assays.

Target ^a^	Herbicide	Population ^b^	Application Rate (g a.i. ha^−1^)
ALS	Imazethapyr	S	0, 0.38, 0.76, 1.52, 3.05, 6.10, 12.18
		R	0, 97.5, 195, 390, 780, 1560, 3120
	Bispyribac-sodium	S	0, 2.81, 5.62, 11.25, 22.5, 45, 90
		R	0, 0.18, 0.35, 0.7, 1.4, 2.8, 5.6
	Florasulam	S	0, 0.02, 0.04, 0.08, 0.16, 0.32, 0.64
		R	0, 4.5, 9, 18, 36, 72, 144
Photosystem II	Bentazone	S	1296, 2592
		R	
	Bromoxynil	S	27, 54
		R	
Synthetic auxin	Fluroxypyr	S	180, 360
		R	
	MCPA	S	1260, 2520
		R	
PPO	Carfentrazone-ethyl	S	562.5, 1125
		R	

^a^ ALS, acetolactate synthase; PPO, protoporphyrinogen oxidase. ^b^ S, susceptible; R, resistant.

## Data Availability

Data are contained within the article.

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
