# Peer review of "First Report of the Molecular Mechanism of Resistance to Tribenuron-Methyl in Silene conoidea L."

_plants, 2022, doi:10.3390/plants11223044_

Round 1

Reviewer 1 Report

The authors present a well-written manuscript showing resistance to TM as well as cross-resistance to other ALS herbicides in Silene conoidea that is conferred by TSR (cross-resistance) and NTSR mechanisms. However, I have some important observations about this manuscript that need to be addressed before publication.

Lines 29-36: As this species is little known in many regions of the world, the authors should include a more detailed botanical description in the manuscript. Such a description of the weed would provide a detailed understanding of its biology and how it has spread in this agricultural region, as well as its evolutionary resistance to the herbicide studied.

Section 2.4 and 4.5: As the authors do not study the metabolic pathways of degradation of the herbicide Tribenuron-methyl. I consider it very important that a dose-response (DR) assay with whole plants is included for a proper comparison of the whole population and not just a single dose. This information is vital to see how much it decreases when comparing these populations (R and S) without and with malathion.

Line 149: Define SU (Don't assume readers know these abbreviations)

Lines 169-171: Should there be a correlation between ALS activity assays and whole plant assays to indicate that there are no additional resistance mechanisms?

Section 4.1: Indicate the growing conditions of the plant from transplanting to use.

Line 193: 10??? Years?

Line: 193: In which growing season were they harvested?

Line 206: Under what greenhouse conditions?

Section 4.3: No indication of the methodology used on how the total ALS activity has been obtained (Table 5).

Section 4.4: Authors must fully describe the PCR conditions used for each primer pair. As this is a manuscript that will be available to everyone, the M&M must be able to be replicated by anyone.

Section 4.6: Under what application conditions were these tests conducted? Under the same conditions as the dose-response assays?

Author Response

Dear reviewer,

Thank you for you precious comments and advice. Those comments are all valuable and very helpful for revising and improving our paper, as well as the important guiding significance to our researches. We have studied comments carefully and have made correction which we hope meet with approval. Below the comments of the reviewers are response point by point and revised portion are marked in red in the paper. We sincerely hope that this revised manuscript has addressed all your comments and suggestions. We appreciated for reviewers’ warm work earnestly, and hope that the correction will meet with approval.

Reviewer 2 Report

The Topic is good and the presentation is generally all right. One thing astonishes me that Results in the precede the Materials and Methods. I have not across this sequence in my whole academic career. If not the format of the journal, the sequence of Results and Materials and Methods may be switched over. Probably it is an omission from the authors because the First Table in Results is Table 4 which does not make sense. these might have gone in chronological order. The subsequent Section i.e Materials and Methods lists Tables 1-3. Hence, rectification is desired please.

My suggestions concerning the ms are attached herewith. Authors are suggested to incorporate these to improve the manuscript. I conclusively categorize this ms as a nice contribution to the Field Weed Science.

 Best Wishes

Author Response

(The authors gave the same response as above.)

Reviewer 3 Report

Even though the current manuscript has novelty in terms of species, however, the methodology and results are not very adequate. Several editings are required prior to publication. Please find the major comments in the attached PDF.

Author Response

(The authors gave the same response as above.)

Round 2

Reviewer 1 Report

Good work, although I still find the introduction too short, the authors could include more data of general interest. But they have adequately integrated my previous remarks.

Regards

Author Response

Dear reviewer,

Thank you for you precious comments and advice. Those comments are all valuable and very helpful for revising and improving our paper, as well as the important guiding significance to our researches. We have studied comments carefully and have made correction which we hope meet with approval. Below the comments of the reviewers are response point by point and revised portion are marked in red in the paper. We sincerely hope that this revised manuscript has addressed all your comments and suggestions. We appreciated for reviewers’ warm work earnestly, and hope that the correction will meet with approval.

Point 1: Good work, although I still find the introduction too short, the authors could include more data of general interest. But they have adequately integrated my previous remarks.

Response 1: We are extremely grateful to reviewer for your suggestion. We have added some information about Silene conoidea L. as you suggest in the revised manuscript. (Line 29-32)

Reviewer 3 Report

I am satisfied with the changes made by the authors. They had made substantial changes. The manuscript is much improved now. I now recommend it for publication. There are only two minor points, which must be resolved.

Minor comment 1: I recommend minor language editing in the introduction and discussion (especially the introduction section).

Minor comment 2: Please improve figures 1 and 3. Currently, they seem to be screenshots. I recommend the authors to improve the quality of the images.

Author Response

Dear reviewer,

Thank you for you precious comments and advice. Those comments are all valuable and very helpful for revising and improving our paper, as well as the important guiding significance to our researches. We have studied comments carefully and have made correction which we hope meet with approval. Below the comments of the reviewers are response point by point and revised portion are marked in red in the paper. We sincerely hope that this revised manuscript has addressed all your comments and suggestions. We appreciated for reviewers’ warm work earnestly, and hope that the correction will meet with approval.

Point 1: I recommend minor language editing in the introduction and discussion (especially the introduction section)

Response 1: We are extremely grateful to reviewer for your suggestion. We have corrected it as you suggest in the revised manuscript.

Point 2: Please improve figures 1 and 3. Currently, they seem to be screenshots. I recommend the authors to improve the quality of the images.

Response 2: We gratefully appreciate for your suggestion. We have modified the size and sharpness of the images in the revised manuscript. (Figure 1, Figure 2 and Figure 3)
